# Incidence, Treatment and Clinical Impact of Iron Deficiency in Chronic Heart Failure: A Longitudinal Analysis

**DOI:** 10.3390/jcm11092559

**Published:** 2022-05-02

**Authors:** Gema Miñana, Miguel Lorenzo, Antonio Ramirez de Arellano, Sandra Wächter, Rafael de la Espriella, Clara Sastre, Anna Mollar, Eduardo Núñez, Vicent Bodí, Juan Sanchis, Antoni Bayés-Genís, Julio Núñez

**Affiliations:** 1Cardiology Department, Hospital Clínico Universitario de Valencia, Universitat de Valencia, INCLIVA, 46010 Valencia, Spain; gemineta@gmail.com (G.M.); miguel.lorenzo.her@gmail.com (M.L.); rdelaespriella@gmail.com (R.d.l.E.); clarasastre90@hotmail.com (C.S.); annam.mollar@gmail.com (A.M.); enunezb@gmail.com (E.N.); vicente.bodi@uv.es (V.B.); sanchis_juafor@gva.es (J.S.); 2CIBER Cardiovascular, 28029 Madrid, Spain; abayesgenis@gmail.com; 3HEOR, Viforpharma Group, 8152 Zurich, Switzerland; antonio.ramirez@viforpharma.com; 4Medical Department, Viforpharma Group, 8152 Zurich, Switzerland; sandra.waechter@viforpharma.com; 5Cardiology Department and Heart Failure Unit, Hospital Universitari Germans Trias i Pujol, 08916 Badalona, Spain; 6Department of Medicine, Autonomous University of Barcelona, 08193 Barcelona, Spain

**Keywords:** iron deficiency, heart failure, prevalence, treatment

## Abstract

In patients with heart failure (HF), iron deficiency (ID) is a well-recognized therapeutic target; information about its incidence, patterns of iron repletion, and clinical impact is scarce. This single-centre longitudinal cohort study assessed the rates of ID testing and diagnosis in patients with stable HF, patterns of treatment with intravenous iron, and clinical impact of intravenous iron on HF rehospitalization risk. We included 711 consecutive outpatients (4400 visits) with stable chronic HF from 2014 to 2019 (median [interquartile range] visits per patient: 2 [2–7]. ID was defined as serum ferritin <100 µg/L, or 100–299 µg/L with transferrin saturation (TSAT) < 20%. During a median follow-up of 2.20 (1.11–3.78) years, ferritin and TSAT were measured at 2230 (50.7%) and 2183 visits (49.6%), respectively. ID was found at 846 (37.9%) visits, with ferritin and TSAT available (2230/4400), and intravenous iron was administered at 321/4400 (7.3%) visits; 233 (32.8%) patients received intravenous iron during follow-up. After multivariate analyses, iron repletion at any time during follow-up was associated with a lower risk of recurrent HF hospitalization (hazard ratio [HR] = 0.50, 95% confidence interval [CI] = 0.28–0.88; *p* = 0.016). Thus, ID was a frequent finding in patients with HF, and its repletion reduced the risk of recurrent HF hospitalizations.

## 1. Introduction

Iron deficiency (ID) is a common nutritional deficiency in patients with heart failure (HF), with a prevalence ranging from approximately half of the patients with stable chronic HF to up to 80% of patients admitted with acute HF [1,2,3]. In addition, ID is associated with an impaired functional capacity and quality of life, and a worse prognosis [3,4]. The ESC HF guidelines recommend periodically screening for anemia and ID testing in patients with HF and systolic dysfunction (Class I) [5,6]. Currently, ID is a recognized therapeutic target in patients with ID and stable and acute HF with systolic dysfunction [5,6,7]. However, data about the frequency of ID testing and diagnosis rates, treatment patterns, and the association of its treatment with clinical outcome in daily clinical practice, especially in longitudinal scenarios, are scarce [8,9,10]. This study aimed to identify the rates of ID testing, diagnosis, and treatment in outpatients with HF. Additionally, we sought to evaluate the association between ID therapy and the risk of recurrent HF hospitalizations.

## 2. Materials and Methods

We retrospectively studied a cohort of 711 patients followed in the HF unit of a tertiary-care hospital in Spain from 2014 to 2019. HF was diagnosed according to the definition proposed by the guidelines [5,11]. All patients with an HF diagnosis were eligible, except those who did not fulfill the HF criteria, according to the guidelines (*n* = 35), with telephone visits (*n* = 112), patients on dialysis (*n* = 18), and remission for other centers (*n* = 23). Patients across all categories of left ventricular ejection fraction (LVEF) were considered (LVEF ≤ 40%, 41–49%, and ≥50%). Patient demographics, medical history, physical examination, 12-lead electrocardiogram, laboratory tests, echocardiogram, and medical treatment were included in pre-established electronic questionnaires.

The local ethics committee approved the study. The study protocol conformed to the 1975 Declaration of Helsinki (revised in 1983), as reflected by a priori approval from the institution’s human research committee. Patients were not involved in designing and conducting this research.

### 2.1. Iron Deficiency Parameters’ Assessment and Definitions

ID was defined, according to ESC criteria [5], as serum ferritin <100 μg/L (absolute ID) or ferritin 100–299 μg/L and transferrin saturation (TSAT) < 20% (functional ID). Anemia was defined, according to the World Health Organization (WHO) criteria, as hemoglobin <12 g/dL in women and < 13 g/dL in men.

The serum ferritin concentration was measured by an immunoturbidimetric assay on the Olympus AU 5400 system (Beckman Coulter, Brea, CA, USA). Colorimetric methods were used to measure the serum iron concentration and unsaturated iron-binding capacity (UIBC), using the Olympus AU 5400 analyzer (Beckman Coulter). The total iron-binding capacity (TIBC) was determined indirectly using the sum of serum iron concentration and UIBC. TSAT was calculated by dividing the serum iron concentration by TIBC and multiplying by 100.

### 2.2. Intravenous Iron Treatment

We registered all the episodes in which intravenous iron treatment with ferric carboxymaltose (FCM) was administered during the follow-up.

### 2.3. Endpoints

The co-primary endpoints were: (1) the incidence of ID screening and its treatment during the follow-up period; (2) the association between ID treatment and the risk of recurrent HF hospitalizations during the same follow-up period. Death status and HF hospitalizations were assessed by reviewing the electronic medical charts.

### 2.4. Statistical Analysis

Continuous variables were expressed as mean ± standard deviation (SD) or median (interquartile range [IQR]), as appropriate. Discrete variables were summarized as percentages. Baseline characteristics were presented for the entire cohort.

Multilevel survival analysis was implemented for evaluating the association between ID and intravenous FCM treatment, with recurrent HF hospitalizations and mortality as terminal events. The interdependence between repeated measures on the same subject was accounted for by modelling patient ID as a random effect (intercept). A flexible spline-based approach on the log hazard (Royston–Parmar) was used to model the survival distribution of the baseline hazard function. When the omnibus *p*-value for the interaction was greater than 0.05, the main terms of the ID and intravenous iron treatment were reported instead. Because of the longitudinal nature of these data, ID was modelled as time-varying exposures. Intravenous iron treatment was considered as a subject constant variable (“1” if FCM was prescribed at least once during follow-up, and “0” otherwise). All regression models included, as controlling covariates, the baseline values of age (years), gender, type 2 diabetes, ischemic etiology of HF, heart rate, systolic blood pressure, LVEF, hemoglobin, estimated glomerular filtration rate, and amino-terminal pro-brain natriuretic peptide (NT-proBNP). Risk estimates were reported as incidence rate ratio (IRR) and hazard ratio (HR).

A 2-sided *p*-value of <0.05 was the threshold used for significance in all analyses. The analysis was implemented with the MERLIN package within STATA 16.1 (Stata Statistical Software, Release 15 [2017]; StataCorp LP, College Station, TX, USA) [12].

## 3. Results

The mean age was 72 ± 12 years, and 445 (62.6%) patients were men. The baseline characteristics of the overall cohort are presented in Table 1. Most patients exhibited New York Heart Association (NYHA) Class II at the first visit (86.5%), and 37.7%, 14.6%, and 47.6% had an LVEF ≤40%, 41–49%, and ≥50%, respectively. The mean (SD) hemoglobin at baseline was 12.9 ± 2.0 g/dL, and 272 (41.2%) patients met the WHO diagnosis of anemia. The mean (SD) serum ferritin and TSAT values were 213.8 ± 249.7 μg/L and 23.1 ± 18.8%, respectively. The median (IQR) value for NT-proBNP at baseline was 1848 pg/mL (646–4637).

### 3.1. Longitudinal Cohort Characteristics

During a median (IQR) follow-up of 2.20 (1.11–3.78) years, 4400 patient health care visits were registered for 711 patients. The median number of visits per patient was 2 (IQR: 2–8; range: 1–47).

### 3.2. Iron Deficiency Assessment and Diagnosis

*Overall findings.* At the first ambulatory visit, ferritin, TSAT, and hemoglobin were measured in 407 (57.2%), 383 (53.9%), and 660 (92.8%) patients, respectively. At this visit, among those in which ferritin and TSAT were measured [391 out of 711 (55%)], 223 (57.0%) met the criteria for ID. In the longitudinal dataset, ferritin and TSAT were measured in 2230 (50.7%) and 2183 (49.6%) person visits (P-Vs), respectively. The number (and proportion) of observations meeting the criteria for ID (as time-varying exposures) (data available in 2230 P-V) was 846 (37.9% on those visits in which ID was tested). When using the definition for ID at any time during the follow-up (at the patient level), the prevalence of ID was 72.4%. Figure 1 summarizes the overall findings.

*LVEF stratified findings.* At the first ambulatory visit, ferritin, TSAT, and hemoglobin were measured in 159 (59.3%), 152 (56.7%), and 250 (93.3%) patients, respectively, for those with LVEF ≤40%; 64 (61.5%), 61 (58.7%), and 97 (93.3%) patients, respectively, for those with LVEF 41–49%; and 184 (54.3%), 170 (50.1%), and 313 (92.3%) patients, respectively, for those with LVEF ≥50%. At this visit, among those in which ferritin and TSAT were measured, there were no differences in the prevalence of ID; [LVEF ≤ 40%] = 87 (57.6%), [LVEF 41–49%] = 31 (49.2%), and [LVEF ≥ 50%] = 105 (59.3%) met the criteria for ID (*p* = 0.373).

In the longitudinal dataset, ferritin and TSAT were measured in 890 (51.1%), 875 (50.2%), and 947 (54.4%) person visits (P-Vs), respectively, for those with LVEF ≤40%; 386 (51.7%), 381 (51.1%), and 408 (54.7%) patients, respectively, for those with LVEF 41–49%; and 954 (49.9%), 927 (48.5%), and 1036 (54.2%) patients, respectively, for those with LVEF ≥50%. The number (and proportion) of observations meeting the criteria for ID (as time-varying exposures) (data available in 2230 P-V on those visits in which ID was tested) was [LVEF ≤ 40%] = 302 (33.9%), [LVEF 41–49%] = 152 (39.4%), and [LVEF ≥ 50%] = 392 (41.1%). When using the definition for ID at any time during the follow-up (at the patient level), the prevalence of ID was 69.2%, 73.9%, and 74.8%, respectively (*p* = 0.437).

### 3.3. Intravenous Iron Treatment

*Overall findings.* At the baseline visit, 73 (10.3%) patients were treated with intravenous iron. Of the 272 patients who presented with anemia at baseline (*n* = 272), 36 (13.2%) received treatment with intravenous iron, compared with 31 (8.0%) of those with no anemia (*n* = 388) (*p* = 0.028).

During the follow-up period, intravenous iron (as time-varying exposures) was administered at 321 (of 4400: 7.3%) P-Vs. In most of the patients (89.4%) receiving FCM, the administered dose was 1000 mg. When using the indicator for intravenous iron at any time during the follow-up (expressed at the patient level), 233 out of 711 (32.8%) patients were treated. At this level, the numbers (%) of patients that received one, two, or three+ administrations of intravenous iron were 168 (23.6%), 50 (7.0%), and 15 (2.1%), respectively. The pattern of intravenous iron administration was 0.22 per person/year.

Among those patients with anemia diagnosed at any time (2871 P-Vs), 1390 (48.4%) received treatment with intravenous iron, compared with 496 (34.3%) P-Vs of those with no anemia (*p* < 0.001). Figure 2 summarizes the findings.

*LVEF stratified findings.* At the baseline visit, [LVEF ≤ 40%] = 25 (9.3%), [LVEF 41–49%] = 8 (7.7%), and [LVEF ≥ 50%] = 40 (11.8%) patients were treated with intravenous iron (*p* = 0.393). The pattern of intravenous iron administration (per person/year) was not different across three categories: [LVEF ≤ 40%] = 0.21; [LVEF 41–49%] = 0.28; and [LVEF ≥ 50%] = 0.20 (*p* = 0.101).

### 3.4. Adverse Clinical Events

All-cause death was observed in 203 patients (28.6%). The overall mortality rate per person/year was estimated at 0.14 (95% confidence interval [CI] = 0.12–0.16). During follow-up, 709 and 248 all-cause and HF hospitalizations, respectively, were observed. The respective overall rates of all-cause and HF hospitalizations per patient/year were 0.48 (95% CI = 0.45–0.52) and 0.17 (95% CI = 0.15–0.19).

### 3.5. Iron Deficiency and HF Rehospitalizations

The crude incidence rates of total HF rehospitalization in patients with and without ID at baseline were 0.286 and 0.160 per person/year, respectively (IRR = 1.79, 95% CI = 1.26–2.53; *p* = 0.001). The main term of ID (as time-varying exposures) did achieve significance as a predictor of HF readmission (HR = 1.86, 95% CI = 1.24–2.80; *p* = 0.003).

In those who were not treated with intravenous iron, ID (as time-varying exposures) was significantly associated with HF rehospitalization (HR = 2.26, 95% CI = 1.28–4.00; *p* = 0.005). Conversely, in those treated with intravenous iron, ID was neutrally associated with this endpoint (HR = 1.56, 95% CI = 0.87–2.78; *p* = 0.136). However, the interaction between ID (as time-varying exposures) and intravenous iron treatment (indicator for any intravenous iron at follow-up) did not reach statistical significance (*p*-value for interaction = 0.362).

### 3.6. Intravenous Iron and Adverse Clinical Events

After the multivariate analysis, intravenous iron repletion at any time during follow-up was associated with a significant reduction in the risk of total HF hospitalizations (HR = 0.50, 95% CI = 0.28–0.88; *p* = 0.016). In addition, the subgroup analyses revealed that intravenous iron treatment was associated with a non-differential protective effect across most representative subgroups, including sex, anemia, and renal function (Figure 3). Additionally, we did not find a heterogeneous association between iron repletion and the risk of readmission, according to LVEF categories (*p* for interaction = 0.956). The HRs for iron repletion for those with LVEF ≤40%, 41–49%, and ≥50% were 0.56 (CI 95% = 0.24–1.24), 0.50 (CI 95% = 0.14–1.82), and 0.44 (CI 95% = 0.17–1.15), respectively (Figure 3). Under this same multivariate scenario, intravenous iron treatment was not related to mortality risk reduction in the whole sample (HR = 0.54, 95% CI = 0.20–1.40; *p* = 0.191).

## 4. Discussion

In this longitudinal study, which included a consecutive and contemporary cohort of 711 patients with HF, visited in a third-level center HF unit, with a median follow-up of 2 years (4400 visits), we found that (1) ID parameters were tested at about half of visits; (2) ID criteria, according to the ESC definition, were met in about 40% of the visits in which ID was tested; (3) intravenous iron was administered in 7% of the visits, but one out of three patients received intravenous iron treatment at least once during follow-up; (4) ID (as time-varying exposures) was associated with an increased risk of HF hospitalization; (5) intravenous iron repletion was associated with a reduction in HF hospitalizations; and (6) we did not find differences in ID diagnosis and treatment across the LVEF categories. The longitudinal characteristic of this study adds a novel piece of information about the trajectory of iron status and its treatment during the natural history of patients with HF.

### 4.1. ID Testing and Treatment in Daily Clinical Practice

The current HF guidelines recommend that patients with HF should be periodically screened for anemia and ID, including during hospitalizations or early after discharge [6]. However, observational data show that ID assessment is far from optimal. For instance, in a large study that included 21,496 HF patients from the Swedish Heart Failure registry in 2017 and 2018 (SwedeHF), Becher et al. reported that only 27% were tested for ID, with slightly higher rates in those with heart failure with a reduced ejection fraction (HFrEF) [8]. The authors also noted that among those in which ID was tested, only 20% of the patients received FCM. These figures seem remarkably low, given that the prevalence of ID in stable ambulatory HF is approximately 50% and even higher in patients with acute HF [2,3]. Similarly, a similar assessment of ID diagnosis and treatment has been reported in other European registries. For instance, a study including data for 2822 acute and chronic HF patients from the *‘Observatoire Français de l’Insuffisance Cardiaque et du Sel*’ (OFICSel) registry in 2017 showed that ID assessment had been performed in only 38.1% of patients [13].

Similarly, a contemporary multicenter Spanish registry of acute HF (*n* = 3555) showed that less than 50% of the patients had been tested for ID during hospitalization [10]. In the current study, we found that, during a median follow-up of 2 years, ID assessment was performed in about 75% of patients and at one of two ambulatory visits. These figures, close to the international recommendations, probably reflect HF management in specialized units. In such units, some data show that ID testing and treatment are superior to that in other non-specialized contexts [8]. In summary, real-world data show that ID diagnosis and treatment remain suboptimal [9]. To improve this situation, we should incorporate TSAT and ferritin in the routine laboratory assessments of patients with HF, and facilitate access to infusion rooms, especially in non-specialized scenarios. For these purposes, proactive institutional initiatives to increase the awareness of ID screening and treatment in HF patients, beyond anemia, seem pertinent [8,9].

### 4.2. ID and Its Treatment and Adverse Clinical Events: Evidence from Real-World Data

Clinical trials have consistently shown that FCM leads to a reduction in the risk of HF readmission [7,14]. Moreover, in a meta-analysis that included 2166 patients from seven randomized clinical trials, treatment with intravenous iron was associated with a lower risk of the composite of HF hospitalization or cardiovascular mortality (odds ratio = 0.73; [95% CI = 0.59–0.90]; *p* = 0.003), with this benefit being driven by the reduction in HF hospitalizations [15]. However, data on the prognostic impact of ID and iron supplementation in real-world clinical practice are scarce. Historical studies have shown that ID identified a subset of HFrEF patients at a higher risk of adverse events, including mortality. However, more recent studies show that there is a positive association between ID and the risk of HF readmission. In patients with chronic HF, the SwedeHF registry reported that ID was associated with a higher risk of recurrent all-cause hospitalizations in HF patients, regardless of anemia [8]. Similar findings have also been reported for patients admitted with acute HF, in which surrogates of ID were related to a higher risk of short-term hospitalization [4,16]. In a retrospective cohort of 565 consecutive outpatients with HF and ID treated with FCM, López-Vilella et al. described a significant improvement in NYHA functional class and surrogates of myocardial function [17].

However, most previous studies have only evaluated one-time ID assessment or treatment. Monitoring ID over time is crucial because ID is a therapeutic target and may be strongly modified by iron supplementation. In the current study, we longitudinally evaluated the prognostic implications of ID and FCM treatment in the daily practice of a specialized HF unit. In our opinion, the current work constitutes a step forward in methodology. In agreement with the findings of randomized clinical trials, we found that ID, as time-varying exposures, was associated with an increased risk of HF hospitalizations. This risk was mitigated when patients were treated with intravenous FCM.

### 4.3. Limitations

This is an observational and retrospective study in which several unmeasured variables might have had a role as confounders. For instance, we did not assess oral iron administration or changes in HF treatments or LVEF during the follow-up. Selection bias cannot be excluded for those patients with ID measurement and treatment. The data reported herein were extracted from the clinical practice records of a third-level HF unit and might not apply to other non-specialized scenarios. In this study, we did not evaluate the association between iron repletion and quality of life, or other surrogates of HF severity.

## 5. Conclusions

In this longitudinal analysis, ID testing and intravenous iron treatment were performed in approximately 50% and 7% of the ambulatory visits in a specialized HF unit, respectively. Intravenous iron repletion at any time during the follow-up occurred in one-third of the patients. Intravenous iron was associated with a lower risk of HF hospitalizations.

## Figures and Tables

**Figure 1 jcm-11-02559-f001:**
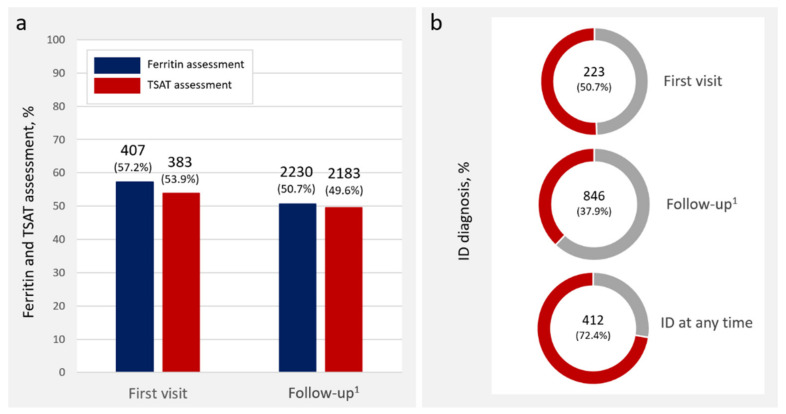
Number and proportion of patients with ID assessment and ID diagnosis: (**a**) Ferritin and TSAT assessment; (**b**) ID diagnosis. ^1^ Person visits. ID: iron deficiency; TSAT: transferrin saturation.

**Figure 2 jcm-11-02559-f002:**
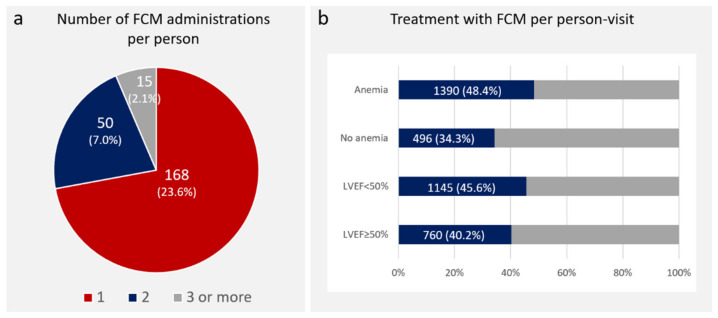
Treatment with FCM. (**a**) Number of FCM administrations of iron per person during the follow-up period; (**b**) rates of FCM administration per person visits, depending on the presence of anemia and among LVEF subgroups. FCM: ferric carboxymaltose; LVEF: left ventricle ejection fraction.

**Figure 3 jcm-11-02559-f003:**
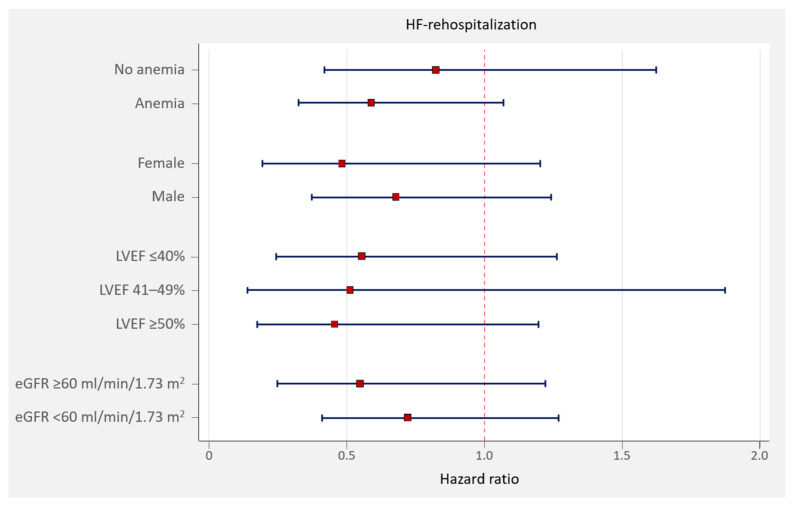
Subgroup analysis. Units for eGFR values are mL/min/1.73 m^2^. Error bars represent 95% confidence intervals. eGFR: estimated glomerular filtration rate; HF: heart failure; LVEF: left ventricle ejection fraction.

**Table 1 jcm-11-02559-t001:** Baseline characteristics.

	All Patients (*n* = 711)
** *Demographics and medical history* **	
Age, years	72 ± 12
Male, *n* (%)	445 (62.6)
Hypertension, *n* (%)	554 (77.9)
Dyslipidaemia, *n* (%)	444 (62.4)
Diabetes, *n* (%)	266 (37.4)
Insulin-dependent diabetes, *n* (%)	86 (12.1)
Active smoker, *n* (%)	98 (13.7)
Previous smoker, *n* (%)	207 (29.1)
Alcohol abuse, *n* (%)	27 (3.8)
Ischaemic heart disease, *n* (%)	318 (44.7)
Valve heart disease, *n* (%)	199 (28.0)
Pacemaker, *n* (%)	36 (5.0)
Implantable cardioverter-defibrillator, *n* (%)	41 (5.8)
Prior stroke, *n* (%)	56 (7.9)
Chronic obstructive pulmonary disease, *n* (%)	115 (16.2)
History of chronic renal disease, *n* (%)	149 (20.9)
History of peripheral artery disease, *n* (%)	52 (7.3)
Prior heart failure hospitalization in the last year, *n* (%)	315 (44.3)
** *Physical examination* **	
Heart rate, bpm	77 ± 17
Systolic blood pressure, mmHg	129 ± 23
Diastolic blood pressure, mmHg	68 ± 13
Pleural effusion, *n* (%)	100 (14.0)
Peripheral oedema, *n* (%)	210 (29.5)
NYHA class, *n* (%)	
I	55 (7.8)
II	615 (86.5)
III	40 (5.6)
IV	1 (0.1)
** *ECG and echocardiography* **	
Left bundle branch block, *n* (%)	190 (26.7)
Atrial fibrillation, *n* (%)	297 (41.8)
Left ventricle ejection fraction, %	47 ± 16
LVEF ≤40%, *n* (%)	268 (37.7)
LVEF 41–49%, *n* (%)	104 (14.6)
LVEF ≥50%, *n* (%)	339 (47.7)
Mitral insufficiency, *n* (%)	90 (12.6)
III/IV Tricuspid insufficiency, *n* (%)	58 (8.2)
TAPSE, mm	18.5 ± 4.2
** *Laboratory data* **	
Haemoglobin, g/dL ^1^	12.9 ± 2.0
Haematocrit, % ^1^	40.1 (6.8)
Anemia (WHO criteria), *n* (%) ^1^	272 (41.2)
Ferritin, μg/L ^2^	213.8 ± 249.7
TSAT, % ^3^	23.1 ± 18.8
Iron deficiency (combined criteria), *n* (%) ^4^	223 (57.0)
Absolute iron deficiency, *n* (%) ^4^	155 (39.7)
Functional iron deficiency, *n* (%) ^4^	68 (17.4)
Urea, mg/dL ^1^	71.3 ± 76.8
Creatinine, mg/dL ^1^	1.31 ± 0.68
eGFR (MDRD formula), mL/min/1.73 m^2 4^	63.7 ± 27.6
eGFR <60 mL/min/1.73 m^2^, *n* (%) ^1^	323 (48.9)
Sodium, mEq/L ^1^	140 ± 3
Potassium, mEq/L ^1^	4.4 ± 0.6
NT-proBNP, pg/mL, median (IQR) ^5,6^	1848 (646–4637)
Carbohydrate antigen 125, U/mL, median (IQR) ^5,6^	23 (12–75)
** *Treatment* **	
Loop diuretics, *n* (%)	587 (82.6)
ACEI or ARB, *n* (%)	463 (51.3)
ARNI, *n* (%)	112 (15.8)
Betablockers, *n* (%)	552 (77.6)
MRA, *n* (%)	365 (51.3)
SGLT2i, *n* (%)	85 (11.9)
Oral anticoagulants, *n* (%)	274 (38.5)
Antiplatelet, *n* (%)	206 (29.0)
Statins, *n* (%)	429 (60.3)
ICD, *n* (%)	41 (5.8)
CRT, *n* (%)	35 (4.9)

Continuous values are expressed as mean ± standard deviation unless otherwise stated. ACEI: angiotensin-converting enzyme; ARB: angiotensin receptor blocker; ARNI: angiotensin receptor/neprilysin inhibitors; CRT: cardiac resynchronization therapy; eGFR: estimated glomerular filtration rate; ICD: implantable cardioverter defibrillator; IQR, interquartile range; LVEF: left ventricular ejection fraction; MDRD: modification of diet in renal disease; MRA: mineralocorticoid receptor antagonist; NT-proBNP: amino-terminal pro-brain natriuretic peptide; NYHA: New York Heart Association; SGLT2i: sodium-glucose cotransporter 2 inhibitors; TAPSE: tricuspid annular displacement systolic excursion; TSAT: transferrin saturation; WHO: World Health Organization. ^1^ Value expressed as median (interquartile range). ^2^ Data available in 660 (92.8%) patients. ^3^ Data available in 407 (57.2%) patients. ^4^ Data available in 383 (53.9%) patients. ^5^ Data available in 391 (55%) patients. ^6^ Data available in 474 (66.7%) patients.

## Data Availability

Data supporting reported results may be provided upon reasonable request.

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
