# Peer review of "Incidence, Treatment and Clinical Impact of Iron Deficiency in Chronic Heart Failure: A Longitudinal Analysis"

_jcm, 2022, doi:10.3390/jcm11092559_

Round 1

Reviewer 1 Report

The manuscript by Gema Miñana et al. entitled “Incidence, treatment and clinical impact of iron deficiency in chronic heart failure: a longitudinal analysis” aimed to identify the rates of iron deficiency testing, diagnosis, and treatment in outpatients with HF. Additionally, they sought to evaluate the association between iron deficiency therapy and the risk of recurrent HF hospitalizations. 
The article is well written and leads some evidence to such point; however, some major issues need to be addressed to improve the significance and reliability of the results of the study:
-Firstly, the exclusion and inclusion criteria are not clear. Please, clarify them in the Methods.
-It would be essential to report in Table 1 the different classes of drugs taken by patients. For example, were there any patients taking anticoagulants or erythropoietin? Moreover, is it possible to exclude oral iron intake in the different groups?
-Can therapy changes be ruled out during follow-up? For example, doctors could have added an ARNI for some patients?
-It would be crucial to report echocardiographic parameters of RV function, such as TAPSE. So, please, add these parameters in Table 1. 
- How many hospitalizations had the different groups of patients had in the previous year?
-Finally, it is important to know if patients who have implanted telemonitoring devices such as Cardiomems, CCM or other devices such as CRT were excluded during the follow-up.As a result, the Reviewer suggests reconsidering the article after major revision.

Author Response

The manuscript by Gema Miñana et al. entitled "Incidence, treatment and clinical impact of iron deficiency in chronic heart failure: a longitudinal analysis" aimed to identify the rates of iron deficiency testing, diagnosis, and treatment in outpatients with HF. Additionally, they sought to evaluate the association between iron deficiency therapy and the risk of recurrent HF hospitalizations. The article is well written and leads some evidence to such point; however, some major issues need to be addressed to improve the significance and reliability of the results of the study:

- Response: We acknowledge this reviewer the positive feedback.-Firstly, the exclusion and inclusion criteria are not clear. Please, clarify them in the Methods.

- Response: Mainly all patients physically visited in the Heart Failure Unit of our center were evaluated (n=711 patients). Those who were telephonically visited, did not fulfill ESC criteria for HF diagnosis, patients on dialysis, and remissions from other centers were excluded. This information was included in the revised version of the manuscript.

- Page 2, lines 51-55, Materials and Methods:

"We retrospectively studied a cohort of 711 patients followed in the HF Unit of a tertiary-care hospital in Spain from 2014 to 2019. HF was diagnosed according to the definition proposed by guidelines [5,11]. All patients with HF diagnosis were eligible, except those who did not fulfill the HF criteria according to guidelines (n=35), with telephone visits (n=112), patients on dialysis (n=18), and remission for other centers (n=23). "

-It would be essential to report in Table 1 the different classes of drugs taken by patients. For example, were there any patients taking anticoagulants or erythropoietin? Moreover, is it possible to exclude oral iron intake in the different groups?

- Response: In the revised table 1, we included the list of the most important treatments at baseline.

Table 1. Baseline characteristics (fragment):

Continuous values are expressed as mean ± standard deviation unless otherwise stated.

ACEI: angiotensin-converting enzyme; ARB: angiotensin receptor blocker; ARNI: angiotensin receptor/neprilysin inhbitors; CRT: cardiac resinchronization therapy; eGFR: estimated glomerular filtration rate; ICD: implantable cardioverter defibrillator; IQR, interquartile range; LVEF: left ventricular ejection fraction; MDRD: Modification of Diet in Renal Disease; MRA: mineralocorticoid receptor antagonist; NT-proBNP: amino-terminal pro-brain natriuretic peptide; NYHA: New York Heart Association; SGLT2i: sodium-glucose cotransporter 2 inhibitors; TSAT: transferrin saturation; WHO: World Health Organization.

-Can therapy changes be ruled out during follow-up? For example, doctors could have added an ARNI for some patients?

- Response: This is a longitudinal study focused on characterizing the trajectory of iron status and iron repletion during a median follow-up of 2 years. Unfortunately, we did not register all pharmacological treatment changes that occurred during the follow-up. It is an arduous task to do. Likewise, we did not register the administration of oral iron. Our protocols do not recommend prescribing oral iron in HF patients; however, patients may receive it from other physicians. Information about this was highlighted as a limitation of the current findings.

-Page 10, 299-301:

"4.3. Limitations

This is an observational and retrospective study in which several unmeasured variables might have had a role as confounders. For instance, we did not assess oral iron administration or changes in HF treatments or LVEF during the follow-up."-It would be crucial to report echocardiographic parameters of RV function, such as TAPSE. So, please, add these parameters in Table 1. 

- Response: In agreement with your suggestion, we included in the tbale 1 the mean (SD) of TAPSE. Additional RV parameters were not registered.Table 1. Baseline characteristics (fragment):

TAPSE: tricuspid annular displacement systolic excursion;

- How many hospitalizations had the different groups of patients had in the previous year?

- Response: Most of the referrals in our Heart Failure Unit come from a recent hospitalization for acute heart failure. Thus, 315 patients from 711 had a prior HF-admission in the last 12 month. This info was included in the revised table 1.

Table 1. Baseline characteristics (fragment):

-Finally, it is important to know if patients who have implanted telemonitoring devices such as Cardiomems, CCM or other devices such as CRT were excluded during the follow-up.

- Response: No patients from this cohort were treated with telemonitoring devices such as Cardiomems or CCM. The number of patients with ICD or CRT at baseline was included in the revised table 1.

Table 1. Baseline characteristics (fragment):

Reviewer 2 Report

In their article Minana et al. presented their original observations regarding ID in patients with chronic HF. The article is well-written, well-structured and provides interesting data. I have only one minor suggestion. As has been mentioned by the authors in the current HF ESC guidelines ID screening and iron supplementation is recommended in HFrEF. However, the authors also considered a significant group of patients with EF>50%. For that reasons, I suggest the presentation of indications for iron supplementation, the clinical characteristics, the results of ID screening in separate paragraphs/tables for HFmrEF and HFpEF patients.

Author Response

In their article Minana et al. presented their original observations regarding ID in patients with chronic HF. The article is well-written, well-structured and provides interesting data. I have only one minor suggestion. As has been mentioned by the authors in the current HF ESC guidelines ID screening and iron supplementation is recommended in HFrEF. However, the authors also considered a significant group of patients with EF>50%. For that reasons, I suggest the presentation of indications for iron supplementation, the clinical characteristics, the results of ID screening in separate paragraphs/tables for HFmrEF and HFpEF patients.

Response: Thanks for your constructive comments. According to your suggestion, most of the presented findings were stratified across the 3 categories of LVEF (≤40%, 41-49%, ≥50%). These new findings were added in the revised version of the manuscript.

Table 1. Baseline characteristics (fragment):

- Page 2, pages 55-57, Materials and methods:

“Patients across all categories of left ventricular ejection fraction (LVEF) were considered (LVEF≤40%, 41-49%, and ≥50%).”

- Pages 5 and 6, pages 144-160, Iron deficiency assessment and diagnosis:

“LVEF stratified findings. At the first ambulatory visit, ferritin, TSAT, and haemoglobin were measured in 159 (59.3%), 152 (56.7%), and 250 (93.3%) patients, respectively for those with LVEF≤ 40%; 64 (61.5%), 61 (58.7%), and 97 (93.3%) patients, respectively for those with LVEF 41-49%; and 184 (54.3%), 170 (50.1%), and 313 (92.3%) patients, respectively for those with LVEF≥ 50%. At this visit, among those in which ferritin and TSAT were measured, there were not differences in the prevalence of ID  [LVEF≤ 40%]=87 (57.6%), [LVEF 41-49%]=31 (49.2%), and [LVEF≥ 50%]=105 (59.3%) met the criteria for ID (p=0.373).

In the longitudinal dataset, ferritin and TSAT were measured in 890 (51.1%), 875 (50.2%), and 947 (54.4%) person-visits (P-Vs), respectively for those with LVEF≤ 40%; 386 (51.7%), 381 (51.1%), and 408 (54.7%) patients, respectively for those with LVEF 41-49%; and 954 (49.9%), 927 (48.5%), and 1,036 (54.2%) patients, respectively for those with LVEF≥ 50%. The number (and proportion) of observations meeting the criteria for ID (as time-varying exposures) (data available in 2,230 P-V on those visits in which ID was tested) was [LVEF≤ 40%]=302 (33.9%), [LVEF 41-49%]=152 (39.4%), and [LVEF ≥ 50%]=392 (41.1%). When using the definition for ID at any time during the follow-up (at patient level), the prevalence of ID was 69.2%, 73.9%, and 74.8%, respectively (p=0.437).” 

- Pages 6 and 7, lines 182-186, Intravenous iron treatment:

“LVEF stratified findings. At the baseline visit, [LVEF≤ 40%]=25 (9.3%), [LVEF 41-49%]=8 (7.7%), and [LVEF≥ 50%]=40 (11.8%) patients were treated with intravenous iron (p = 0.393). The pattern of intravenous iron administration (per person-year) was not different across 3 categories [LVEF≤ 40%]=0.21; [LVEF 41-49%]=0.28, and [LVEF≥ 50%]=0.20 (p = 0.101).”

- Page 8, page 220-223, Intravenous iron and adverse clinical events:

“Additionally, we did not find a heterogeneous association between iron repetition and risk of readmission according to LVEF categories (p-for interaction=0.956). The HRs for iron repletion for those with LVEF≤40%, 41-49%, and ≥50% were 0.56 (CI 95%=0.24-1.24), 0.50 (CI 95%=0.14-1.82),and 0.44 (CI 95%=0.17-1.15), respectively (Figure 3).”

- Page2 8 and 9, lines 233-244, Discussion:

“In this longitudinal study, which included a consecutive and contemporary cohort of 711 patients with HF visited in a third-level centre HF-unit with a median follow-up of 2 years (4,400 visits), we found that: (1) ID parameters were tested at about half of visits; (2) ID criteria, according to the ESC definition, were met in about 40% of the visits in which ID was tested; (3) intravenous iron was administered in 7% of the visits, but 1 of 3 patients received intravenous iron treatment at least once during follow-up; (4) ID (as time-varying exposures) was associated with an increased risk of HF hospitalization;  (5) intravenous iron repletion was associated with a reduction in HF hospitalizations, and (6) we did not find differences in ID diagnosis and treatment across LVEF categories. The longitudinal characteristic of this study adds a novel piece of information about the trajectory of iron status and its treatment during the natural history of patients with HF.”

- Figure 3 was modified to include the three categories of LVEF:

Figure 3:

Reviewer 3 Report

The paper is well written. However, I do not understand what is new and interesting regarding this subject. Iron Def. in HF patients was and is widely investigated.

It is relevant but it is obvious and it is not new.

The idea is not original and it does not add new information.

The paper is well written and the text is clear.

The conclusions are clear and correct.

Author Response

The paper is well written. However, I do not understand what is new and interesting regarding this subject. Iron Def. in HF patients was and is widely investigated. It is relevant but it is obvious and it is not new. The idea is not original and it does not add new information. The paper is well written and the text is clear. The conclusions are clear and correct.

- Response: We acknowledge this reviewer the comments. Regarding what novel information is providing the current findings, we want to point out that, conversely to most of the literature, we have evaluated the trajectory of iron status and the long-term prognostic implications of time-updated ID and iron repletion in daily clinical practice. Most of the published literature has assessed the clinical implications of ID and iron repletion in non-longitudinal designs, ignoring that ID status may change over time. According to the reviewer's comment, we emphasize the longitudinal characteristics of this study design.

- Page, lines 253-255, Discussion:

“In this longitudinal study, which included a consecutive and contemporary cohort of 711 patients with HF visited in a third-level centre HF-unit with a median follow-up of 2 years (4,400 visits), we found that: (1) ID parameters were tested at about half of visits; (2) ID criteria, according to the ESC definition, were met in about 40% of the visits in which ID was tested; (3) intravenous iron was administered in 7% of the visits, but 1 of 3 patients received intravenous iron treatment at least once during follow-up; (4) ID (as time-varying exposures) was associated with an increased risk of HF hospitalization;  (5) intravenous iron repletion was associated with a reduction in HF hospitalizations, and (6) we did not find differences in ID diagnosis and treatment across LVEF categories. The longitudinal characteristic of this study adds a novel piece of information about the trajectory of iron status and its treatment during the natural history of patients with HF.”

Additionally, we expanded the limitations section, highlighting the inherent limitation of this type of observational study.

- Page 10, lines 299-301, Limitations:

“This is an observational and retrospective study in which several unmeasured variables might have had a role as confounders. For instance, we did not assess oral iron administration or changes in HF treatments or LVEF during the follow-up.”

Round 2

Reviewer 1 Report

The authors did not register all pharmacological treatment changes and the administration of oral iron that occurred during the follow-up. This point is crucial. 

Author Response

In this revised version, we included pharmacological and devices treatment at baseline. However, as has been addressed in the limitation section, we did not register changes in HF pharmacological treatment or oral iron occurred during follow-up. We are aware that it is a major limitation.

However, we believe this limitation did not invalidate current findings for the following reasons.

  1. Oral iron has failed to show a clinical benefit in HF patients (JAMA. 2017;317(19):1958-1966).
  2. Patients here included were on a high proportion of guideline medical therapy as is shown in table 1 (about 70% RAASi, 80 B-blockers, 50% MRA). Additionally, there were a high proportion of patients with HFpEF, an entity in which optimal pharmacological treatment remains more elusive.

Reviewer 3 Report

Good job

Author Response

Thank you